# Non-Surgical Periodontal Treatment Impact on Subgingival Microbiome and Intra-Oral Halitosis

**DOI:** 10.3390/ijms24032518

**Published:** 2023-01-28

**Authors:** Catarina Izidoro, João Botelho, Vanessa Machado, Ana Mafalda Reis, Luís Proença, Helena Barroso, Ricardo Alves, José João Mendes

**Affiliations:** 1Periodontology Department, Egas Moniz Center for Interdisciplinary Research, Egas Moniz School of Health & Science, 2829-511 Almada, Portugal; 2Clinical Research Unit (CRU), Egas Moniz Center for Interdisciplinary Research, Egas Moniz School of Health & Science, 2829-511 Almada, Portugal; 3Instituto de Ciências Biomédicas Abel Salazar, School of Health and Life Sciences, University of Porto, 4099-002 Porto, Portugal; 4Neuroradiology Department, Hospital Pedro Hispano, 4464-513 Matosinhos, Portugal; 5Quantitative Methods for Health Research Unit (MQIS), Egas Moniz Center for Interdisciplinary Research, Egas Moniz School of Health & Science, 2829-511 Almada, Portugal; 6Microbiology and Public Health Unit, Egas Moniz Center for Interdisciplinary Research, Egas Moniz School of Health & Science, 2829-511 Almada, Portugal

**Keywords:** halitosis, subgingival microbiome, periodontitis, periodontal disease, periodontal medicine, volatile sulfurous compounds

## Abstract

The purpose of this study was to characterize and compare subgingival microbiome before and after periodontal treatment to learn if any changes of the subgingival microbiome were reflected in intra-oral halitosis. We tested the hypothesis that intra-oral halitosis (Volatile sulfur compounds levels) correlates with corresponding subgingival bacterial levels before and after periodontal treatment. Twenty patients with generalized periodontitis completed the study. Subgingival plaque samples were collected at baseline and 6–8 weeks after nonsurgical periodontal therapy. Full-mouth periodontal status assessed probing depth (PD), clinical attachment loss (CAL), gingival recession (REC), bleeding on probing (BoP), PISA and PESA. Halitosis assessment was made using a volatile sulfur compounds (VSC) detector device. Periodontal measures were regressed across VSC values using adjusted multivariate linear analysis. The subgingival microbiome was characterized by sequencing on an Illumina platform. From a sample of 20 patients referred to periodontal treatment, 70% were females (*n* = 14), with a mean age of 56.6 (±10.3) years; full-mouth records of PD, CAL, BOP (%) allowed to classify the stage and grade of periodontitis, with 45% (*n =* 9) of the sample having Periodontitis Stage IV grade C and 95% (*n* = 19) had generalized periodontitis. The correlation of bacterial variation with VSCs measured in the periodontal diagnosis and in the reassessment after treatment were evaluated. *Fusobacterium nucleatum*, *Capnocytophaga gingivalis* and *Campylobacter showaei* showed correlation with the reduction of VSC after periodontal treatment (*p*-value = 0.044; 0.047 and 0.004, respectively). *Capnocytophaga sputigena* had a significant reverse correlation between VSCs variation from diagnosis (baseline) and after treatment. Microbial diversity was high in the subgingival plaque on periodontitis and intra-oral halitosis participants of the study. Furthermore, there were correlations between subgingival plaque composition and VSC counting after periodontal treatment. The subgingival microbiome can offer important clues in the investigation of the pathogenesis and treatment of halitosis.

## 1. Introduction

Periodontitis is a plaque-induced chronic inflammatory disease affecting the periodontium [1]. It is a biofilm-mediated and multifactorial disease, in which the subgingival microbiome plays a critical role in its onset and progression of the disease [2]. The subgingival niche offers optimal ecological conditions for microbial growth [3,4]. *Porphyromonas gingivalis*, *Tannerella forsythia*, and *Treponema denticola* are considered keystone periodontal pathogens [5], yet other relevant microorganisms are present suggesting its subgingival screening is highly relevant [6,7]. In addition, periodontitis consistently links with systemic health through its pathogenic bacteria, bioproducts and associated low-grade general inflammatory state [8].

Among clinical signs and symptoms, halitosis is a negative consequence of periodontitis [9,10] and contributes to the deteriorating effect on oral health-related quality of life (OHRQoL) [11,12]. Halitosis is characterized by an unpleasant odor emanating from the mouth, either from oral or non-oral sources [13]. About 80–90% of the causes are estimated to derive from the oral cavity [14,15]. Depth pockets act as reservoirs of microorganisms responsible for secreting sulfur components that contribute to this unpleasant smell [10,16]. Volatile sulfur compounds (VSC) can be toxic for human cells even at low concentrations [17]. Several studies demonstrate the importance of studying volatile compounds (VSCs and VOCs) in detecting oral diseases [18].

While observational studies could associate periodontal pathogens to VSC levels production, subgingival microbiome characterization after periodontal treatment and changes to the VSC level is scarce. This may contribute to unveiling whether expected subgingival plaque shift after treatment impacts the production of VSCs. Nonsurgical periodontal treatment (that is, scaling and root planing), induces subgingival ecological and composition changes [19].

To the best of our knowledge, the effect of scaling and root planing on the subgingival microbiome and its outcome on intraoral VSCs is poorly understood. Thus, the purpose of this non-randomized controlled trial was to characterize the subgingival microbiome and quantify the VSC levels before and after periodontal treatment.

## 2. Materials and Methods

### 2.1. Study Design

This non-randomized intervention study recruited participants consecutively from the Periodontology Department of Egas Moniz Dental Clinic (EMDC), between December 2021 and June 2022. This study was approved by the respective Institutional Review Board (Egas Moniz Ethics Committee ID no. 781) and within the Helsinki Declaration of 1975 (as per the 2013 revision). Patients were informed of the aims of the study and following the agreement to participate in the study, signed the written consent form. All clinical examinations and treatments were performed by the same clinician (C.I.).

### 2.2. Participants and Eligibility Criteria

Participants were enrolled if they fulfilled the following inclusion criteria: diagnosis of periodontitis; age between 18 and 65 [20], and ability to provide informed consent. Patients were excluded if they had received treatment for periodontitis in the last 12 months; if they had taken antibiotics in the last month; previous record of head and neck radiotherapy; chemotherapy in the previous 6 months; extra-oral causes of halitosis (for instance, respiratory tract inflammatory conditions, cancer, kidney diseases, diabetes mellitus or antidepressants) [21,22]; pregnancy; systemic medication resulting in hyposalivation; or, a diagnosis of diabetes mellitus.

Without mentioning that the halitosis measurement was planned (to minimize any bias source) at the baseline, participants were informed two days prior to the periodontal assessment [20] to: avoid particular foods (namely, spicy aliments, onions and garlic) 24 to 48 h prior examination; avoid smoking 4 to 12 h before the exam; perform oral hygiene 12 h before, if the exam was performed in the morning, or 4 h before if it was performed in the afternoon; consume water until 1 h before treatment; avoid the use of perfumes and deodorants within 24 h before the test.

### 2.3. Variables

#### 2.3.1. Sociodemographic and Medical Questionnaires

Among the sociodemographic information, we collected sex, age, schooling (no education, elementary, middle or higher), job status (student, employed, unemployed or retired), marital status (single, married, divorced or widowed), smoking habits (amount and length), alcoholic habits (amount and frequency), and average family monthly income (in euros). Among the medical information, participants reported existing systemic diseases, prescriptions, and oral hygiene habits (frequency and devices used). The overall dental observation included presence of: caries, retained roots, fixed prosthesis, removable prosthesis, poorly adapted restorations, supragingival calculus, presence of implants, peri-implantitis, recent extractions and presence of dental abscesses.

#### 2.3.2. Periodontal Assessment

The periodontal diagnosis was based on a circumferentially full-mouth protocol using a manual periodontal CP-12 probe (Hu-Friedy^®^, Chicago, IL, USA). Measurements were made at six sites per tooth (mesiobuccal, buccal, distobuccal, mesiolingual, lingual, and distolingual) [23]. Periodontal pocket depth (PPD) measured the distance from the free gingival margin to the bottom of the pocket and recession from the cementoenamel junction (CEJ) to the free gingival margin. Clinical attachment loss was the result of the sum PPD and recession for each site. Measurements were rounded to the lowest whole millimeter. Furcation involvement (FI) was assessed using a Naber’s probe [24]. Tooth mobility was further appraised [25]. We defined periodontitis according to the AAP/EFP 2018 consensus, that is, if interdental CAL was detectable at ≥2 mm non-adjacent teeth, or buccal or oral CAL ≥ 3 mm with pocketing >3 mm was detectable at ≥2 teeth [26]. We further completed the diagnosis with the staging and grading (see Appendix A).

#### 2.3.3. Halitosis Assessment

The diagnosis of halitosis was carried out in two steps: (1) self-reported questionnaire, to exclude possible causes for extra-oral halitosis (referred in Section 2.2) [21,22]; (2) halitosis assessment through a monitoring VSCs device (Halimeter^®^, Interscan Corp, Chatsworth, CA, USA).

Before measurement, patients kept their mouth closed for 1 min. The end of the cannula was then inserted into the patient’s mouth, and the VSC score recorded at the maximum peak displayed by the device. The result was interpreted according to the manufacturer’s instructions and as previously reported: less than 80 ppb denoted no perceptible odor, 80 to 100 ppb denoted perceivable odor, 100 to 120 ppb denoted moderate halitosis, 120 to 150 ppb denoted more pronounced halitosis, and >150 ppb denoted severe halitosis [27,28].

#### 2.3.4. Treatment Protocol

After completion of baseline monitoring, individuals received scaling and root planing (SRP) in two sessions (one side per session) under local anesthetic and instruction in proper home care procedures. Approximately 2 months after SRP, individuals were re-examined (periodontal clinical parameters; VSC levels; subgingival sample collection) as part of their periodontal maintenance.

#### 2.3.5. Subgingival Microbiome Sequencing

Samples were collected using sterile curettes, suspended directly in Tris-EDTA Buffer, pH 8.0 in a sterile Eppendorf, and stored at −20 °C during a week (short-term) and then at −80 °C (long term) until analysis.

Genomic DNA was extracted according to the manufacturer’s instructions (ExtractMe DNA Tissue Kit, Blirt, Gdańsk, Poland). The extracted DNA was stored eluted in Elution Buffer at −20 °C until sequencing.

Sequencing was performed by Novogene (Cambridge, UK). 16SrRNA genes (16SV3-V4) were amplified. All PCR reactions were carried out with the Phusion^®^ High-Fidelity PCR Master Mix (New England Biolabs, Ipswich, MA, USA). PCR products were mixed with the same volume of 1X loading buffer (contained SYB green)and were separated by electrophoresis on 2% agarose gel for detection. Samples with a bright main strip between 400–450 bp were chosen for purification using a Qiagen Gel Extraction Kit (Qiagen, Germany).

Sequencing libraries were generated using the NEB Next Ultra DNA Library Pre^®^ Kit (New England Biolabs) for Illumina sequencing, following manufacturer’s recommendations and index codes were added during library preparation. The library quality was assessed on the Qubit@ 2.0 Fluorometer (Thermo Scientific, Waltham, MA, USA) and Agilent Bioanalyzer 2100 system. At last, the library was sequenced on an Illumina platform and 250 bp paired-end reads were generated.

Paired-end reads were assigned to samples based on their unique barcodes and truncated by cutting off the barcode and primer sequence, being merged using Flash. Quality filtering on the raw tags were performed under specific filtering conditions to obtain the high-quality clean tags [29] according to the QIIME(v1.7.0) [30] quality controlled process.

Tags were compared with the reference database (Gold database) using the UCHIME algorithm (UCHIME Algorithm) [31] to detect chimera sequences, and these were removed [32]. Then, the Effective Tags were finally obtained.

Sequence analysis was performed using the Uparse software (v7.0.1001) [33]. Sequences with ≥97% similarity were assigned to the same Operational Taxonomic Unit (OTUs). A representative sequence for each OTU was screened for further annotation. For each representative sequence, the GreenGenes Database [34] was used based on the RDP classifier algorithm [35] to annotate taxonomic information. In order to study phylogenetic relationship of different OTUs, and the difference of the dominant species in different samples(groups), multiple sequence alignment were conducted using the MUSCLE software (v 3.8.31) [36]. The abundance of OTUs was normalized using a standard of sequence number corresponding to the sample with the least sequences. Subsequent analysis of alpha diversity and beta diversity were all performed basing on this output normalized data, using QIIME. Cluster analysis was preceded by a principal component analysis (PCA), which was applied to reduce the dimension of the original variables using the FactoMineR package and ggplot2 package in R software(Version 2.15.3). Principal Coordinate Analysis (PCoA) was performed to obtain principal coordinates and visualize the complex, multidimensional data. A distance matrix of weighted or un-weighted unifrac among samples obtained before was transformed to a new set of orthogonal axes, by which the maximum variation factor was demonstrated by the principal coordinate, and the second maximum one by the second principal coordinate, and so on. The unweighted Pair-group Method with Arithmetic Means (UPGMA) clustering was performed as a type of hierarchical clustering method to interpret the distance matrix using average linkage and was conducted by QIIME software (v 1.7.0).

### 2.4. Measurement Reliability and Reproducibility

One examiner (C.I.)was previously calibrated for the periodontal examination in five patients not included in the study. Intra-class correlation coefficient (ICC) was computed to ascertain reliability and reproducibility. For both PPD and CAL, the inter-examiner agreement was 0.97 and 0.94, respectively.

### 2.5. Statistical Analysis

All statistical analyses were conducted in R (v4.1.0). After testing for data normality, clinical data and questionnaire results were compared between baseline and follow-up. The Mann–Whitney test was used in continuous variables. Then, we the explored correlation of baseline (B), follow-up (FW) and the difference between B and FW (ΔB-FW) with the relative abundance levels of known periodontopathogenic bacteria (*P. gingivalis*, *T. forsythia*, *T. denticola*, *F. nucleatum*, *C. showae*, *C. gingivalis*, *C. sputigena*, *P. intermedia*, *P. nigrescens*, *F. periodonticum*, *P. micra*, *C.ochracea*, *A. actinomycetemcomitans*, *A. israelii*, *Actinomyces gerencseriae*). For this purpose, Spearman correlation test was used. A *p*-value < 0.05 was considered statistically significant.

## 3. Results

From a sample of 20 patients referred to periodontal treatment, 70% were females (*n =* 14), with a mean age of 56.6 (±10.3) years; a total of 65% were non-smokers (*n =* 13), 20% (*n =* 4) had reported controlled hypertension (Table 1).With regard to oral hygiene habits, 85% (*n =* 17) of participants were referred to brush their teeth twice a day, with 90% (*n =* 18) using manual toothbrush, 65% (*n =* 13) using mouthwash, and 20% (*n =* 4) using interdental flossing (Table 1).

Full-mouth records of PD, CAL, BOP (%) and PI (%) allowed to classify the stage and grade of periodontitis, with 45% (*n =* 9) of the sample having Periodontitis Stage IV grade C and 95% (*n* = 19) had generalized periodontitis.

VSCs, Winkler Index, OHIP-14 (functional limitation, physical pain, psychological discomfort, physical disability, psychological disability, social disability and handicap); mean of CAL and PPD at Baseline and follow-up (6–8 weeks before periodontal treatment) are presented in Table 2. Conventional periodontal treatment resulted in a decrease in the means of the VSCs, Winkler index and OHIP-14, although without statistically significant differences, in the analyzed sample.

When analyzing the number of species of samples of the baseline and after treatment, we observed a relative maintenance without statistical differences (*p* = 0.9033).

### 3.1. Impact of Periodontal Treatment on Subgingival Microbiome

Relative abundance of the 10 predominant subgingival bacterial taxa phylum before and after nonsurgical periodontal treatment is presented in Figure 1 (per patient) and Figure 2 (overall). The predominant bacterial taxa phylum in subgingival plaque were *Bacteroidota*, *Firmicutes*, *Proteobacteria*, *Fusobacteriota*, *Actinobacteriota* and *Sprirochaetota*, which constituted approx. ninety percent of all DNA reads before and after nonsurgical periodontal therapy (Figure 2). In addition, no significant differences in relative abundance of predominant bacterial taxa phylum were recorded the follow-up period (Figure 2).

The predominant bacterial species in subgingival plaque were *Prevotella*, *Porphiromonas*, *Fusobacterium*, *Treponema*, *Neisseria and Leptotrichia* (Figure 3).

### 3.2. Correlation between Subgingival Microbiome Abundance and VSC before and after Periodontal Treatment

The correlation of bacterial variation with VSCs levels before and after periodontal treatment was evaluated (Table 3). *Fusobacterium nucleatum*, *Capnocytophaga gingivalis* and *Campylobacter showaei* showed correlation with the reduction in VSC after periodontal treatment (*p*-value = 0.044; 0.047 and 0.004, respectively). *Capnocytophaga sputigena* had a significant reverse correlation between VSCs variation from diagnosis (baseline) and after treatment (Table 3).

## 4. Discussion

This interventional study explored the variation of the subgingival microbiome and of the intraoral VSCs among adults with periodontitis, before and after treatment. Overall, an expected disruption of the subgingival microbiome was noticed as a result of mechanical root planing. When correlating these levels with intra-oral VSCs, our results showed that two orange complex bacteria (*F. nucleatum* and *C. showae*) and one green complex bacterium (*C. gingivalis*) had strong correlation with VSCs after therapy. On the contrary, one green complex bacterium (*C. sputigena*) was correlated to lower VSC differences of baseline to follow-up.

These correlations may be relevant because they identify four specific bacteria from the dental plaque of pathological periodontal pockets that correlate to clinical levels of halitosis. Additionally, these results may guide future research on the mechanisms involved in the production of VSCs by these bacteria. On the other hand, novel therapies may be designed to target these specific microorganisms and their VSC production; therefore, resulting in novel approaches for mitigating halitosis of periodontal reason.

The subgingival biofilm of the periodontium is mostly a Gram-negative anaerobic population niche with proteolytic capacity [37]. Biologically, these bacteria are capable of degrading sulfur-containing substrates, including the periodontal pockets, releasing volatile sulfur compounds (VSCs).

There are several mechanisms that can explain the link between halitosis and periodontal disease, usually based on properties of the main microbially generated VSCs, where hydrogen sulfide and methylmercaptan facilitate the penetration of lipopolysaccharide into the gingival epithelium, inducing inflammation [38]. The VSCs also aid bacterial invasion of the connective tissue by their toxic effects on epithelial cells, while methyl mercaptan hinders epithelial cell growth and proliferation [39]. This is accentuated by decreasing oxygen tension arising from an increase in periodontal pocket depth, with a concomitant decrease in pH, which is necessary for the putrefaction of amino acids that create VSCs.

*Fusobacterium nucleatum* is an anaerobic oral commensal and a periodontal pathogen (orange complex) associated with a wide spectrum of human diseases [40]. *F. nucleatum* is one of the most abundant species in the oral cavity, in both diseased and healthy individuals [41,42,43,44]. It is implicated in various forms of periodontal diseases including the mild reversible form of gingivitis and the advanced irreversible forms of periodontitis [42,43,44,45,46]. The prevalence of *F. nucleatum* increases with the severity of disease, progression of inflammation and pocket depth [42,45,46].

In this study, we observed that the reduction in this bacterial species after periodontal treatment showed a strong correlation with the reduction in VSCs, which is confirmed by other authors, who associate this bacteria with the production of hydrogen sulfide [47].

*Campylobacter showaeis*, a bacterium historically linked to gingivitis and periodontitis (orange complex), has recently been associated with inflammatory bowel disease and colorectal cancer [48]. This bacterial species correlated significantly with VSC reduction after treatment. Additionally, the microbiome composition of the tongue microbiome was reported to present *Aggregatibacter*, *Campylobacter*, *Capnocytophaga*, *Clostridiales*, *Leptotrichia*, *Parvimonas*, *Peptostreptococcus*, *Peptococcus*, *Prevotella*, *Selenomonas*, *Dialister*, *Tannerella*, and *Treponema bacteria* in the group of patients with IOH [49].

*Capnocytophaga* gingivalis(green complex) is a facultative anaerobic, capnophilic, fusiform, Gram-negative bacilli exhibiting gliding motility [50]. This species is saccharolytic and manifests an increased biomass and proteolytic potential when grown in elevated glucose conditions [51,52].

Low VSC levels present cellular toxicity potential in human cells. They contain thiols (-SH groups) that interact with other proteins and support the negative interaction of bacterial antigens and enzymes. The result of this effect is chronic inflammation, periodontal gingivitis, and periodontitis [17]. In human gingival fibroblasts, H_2_S induces mitochondrial apoptosis [53] and is a known genotoxic agent, with impact on genomic instability and cumulative mutations [54]. In preclinical animal studies, hydrogen sulfide led to ultrastructural changes in epithelial cells and periodontal destruction [55]. Other consequences may result from high levels of H_2_S such as activation of proliferation, migration, and invasion that may lead to carcinogenesis [56,57].

Several species have been strongly correlated with oral dysbiosis and oral carcinoma, such as *Capnocytophaga gingivalis*, *Fusobacterium* sp., among others, due to the fact that these bacteria may promote inflammation, cell proliferation and the production of some oncogenic substances, [58] and VSCs may be involved in this process.

### Strengths and Limitations

Some limitations apply to this investigation, including the withdrawal of three patients who did not attend the reassessment visit 6 to 8 weeks after periodontal treatment.

As mentioned, oral hygiene and feeding instructions were given before the VSC measurement. However, we cannot guarantee that participants strictly adhered to the given recommendations. We consider that this may be a limitation of the study, but that it seems to be common to most clinical studies involving the evaluation of halitosis.

For practical and economical reasons, analysis of a pooled subgingival sample from the four deepest periodontal lesions was used. However, as all participants had more than the four diseased sites from where samples were taken, information on the complete subgingival microbiome was not obtained. This limitation highlights the major dilemma of using local microbial sampling in clinical periodontology, namely that ideally single-site sampling analysis should be performed. However, this may not always be practically feasible.

## 5. Conclusions

Microbial diversity was high in the subgingival plaque on periodontitis and intra-oral halitosis participants of the study. Furthermore, there were correlations between subgingival plaque composition and VSC quantification after periodontal treatment. The subgingival microbiome can offer important clues in the investigation of the pathogenesis and treatment of halitosis.

## Figures and Tables

**Figure 1 ijms-24-02518-f001:**
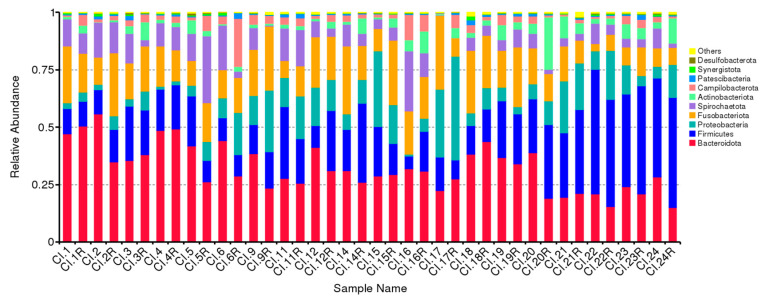
Predominant bacterial taxa phylum in subgingival plaque. Each variable of the Sample Name is named without R (baseline) and with R (follow-up).

**Figure 2 ijms-24-02518-f002:**
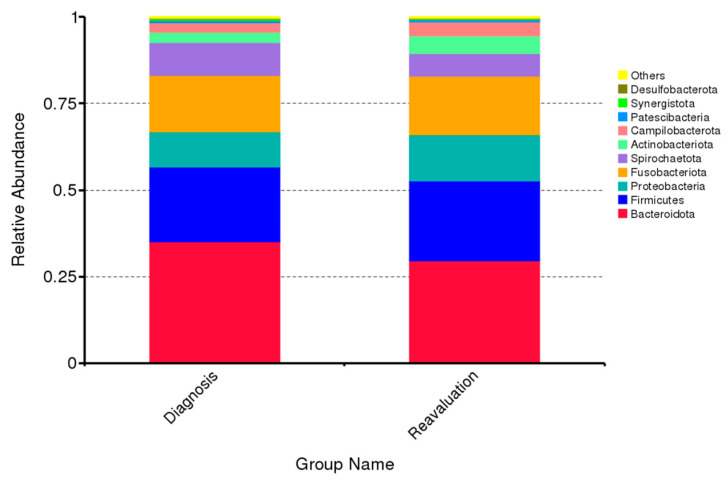
Mean levels of relative abundance of the 10 predominant phyla in subgingival samples at baseline and 6–8 weeks after treatment.

**Figure 3 ijms-24-02518-f003:**
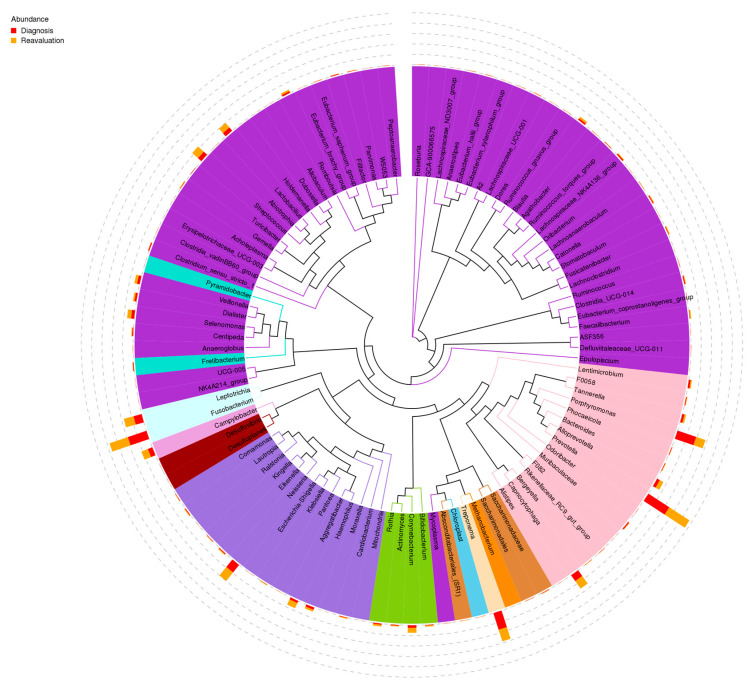
Predominant bacterial species in subgingival plaque. Mean levels of relative abundance of the 100 predominant species in subgingival samples at baseline and 6–8 weeks after treatment.

**Table 1 ijms-24-02518-t001:** Participants baseline sociodemographic and clinical characteristics.

Variable	
Age, mean (SD) (min-max)	56.6 (10.3) (41–80)
Females, % (*n*)	70.0 (14)
Systemic conditions, % (*n*)	
Hypertension	20.0 (4)
Arthritis	5.0 (1)
Toothbrushing per day, % (*n*)	
1	15.0 (3)
2	85.0 (17)
Oral hygiene habits, % (*n*)	
Manual brush	90.0 (18)
Tongue scraper	0.0 (0)
Mouthwash	65.0 (13)
Interdental flossing	20.0 (4)
Interdental brush	25.0 (5)
Denture	15.0 (3)
Smoking habits, % (*n*)	
Never	65.0 (13)
Active	35.0 (7)
Alcohol consumption, % (*n*)	60.0 (12)
Periodontitis staging and grading, % (*n*)	
II-C	5.0 (1)
III-B	35.0 (7)
III-C	5.0 (1)
IV-B	10.0 (2)
IV-C	45.0 (9)
Generalized periodontitis	95.0 (19)

**Table 2 ijms-24-02518-t002:** Participants baseline and follow-up periodontal and halitosis measurements.

Variable	Baseline	Follow-Up	*p*-Value *
VSCs, mean (SD) (min-max)	115.2 (113.7)	58.0 (52.3)	0.007
Winkler Index, median (min-max)	3.0 (0–8)	2.0 (0.8)	0.231
OHIP-14, mean (SD)			
Total score	18.7 (11.4)	17.8 (11.4)	0.588
Functional limitation	1.9 (1.7)	2.2 (1.7)	0.277
Physical pain	3.5 (2.0)	3.4 (2.0)	0.937
Psychological discomfort	4.4 (2.5)	4.1 (2.2)	0.546
Physical disability	3.6 (2.4)	3.2 (2.1)	0.382
Psychological disability	2.7 (2.3)	2.2 (2.2)	0.253
Social disability	1.9 (2.3)	1.8 (2.4)	0.774
Handicap	1.9 (1.9)	1.9 (2.4)	0.863
Mean CAL, mean (SD)	3.3 (0.8)	2.6 (0.6)	0.457
Mean PPD, mean (SD)	4.0 (1.7)	3.8 (1.6)	0.406
Distribution of PPD > 5 mm, % (*n*)	18.8 (15.9)	4.8 (10.2)	0.368
Distribution of CAL > 7 mm, % (*n*)	14.5 (20.3)	10.8 (18.9)	0.194

* Mann–Whitney test for continuous variables.

**Table 3 ijms-24-02518-t003:** Correlation of percentage microbiome with VSCs variation.

Complex	Bacteria	VSC Levels					
		ΔB-FW	*p*-Value	B	*p*-Value	FW	*p*-Value
Red	*P. gingivalis*	0.044	0.855	−0.160	0.500	−0.261	0.267
	*T. forsythia*	−0.254	0.281	0.207	0.382	−0.056	0.816
	*T. denticola*	0.013	0.957	−0.006	0.980	0.013	0.960
Orange	*F. nucleatum*	0.047	0.844	0.166	0.483	0.456	**0.044**
	*C. showae*	−0.744	0.149	0.824	0.086+	0.976	**0.004**
Green	*C. gingivalis*	0.062	0.795	0.149	0.530	0.449	**0.047**
	*C.sputigena*	−0.451	**0.046**	0.281	0.229	−0.287	0.219
	*P. intermedia*	0.130	0.584	−0.120	0.613	−0.002	0.994
	*P. nigrescens*	0.342	0.141	−0.247	0.293	0.143	0.548
	*F. periodonticum*	0.014	0.955	0.112	0.638	0.271	0.248
*-*	*P. micra*	0.094	0.694	−0.138	0.563	−0.112	0.637
	*C. ochracea*	0.099	0.678	−0.042	0.860	0.106	0.657
	*A. actinomycetemcomitans*	−0.232	0.520	−0.023	0.949	−0.265	0.460
	*A. israelii*	−0.200	0.427	0.109	0.666	−0.163	0.517
	*Actinomyces gerencseriae*	0.363	0.139	−0.184	0.465	0.326	0.187

Abbreviations: B: baseline; FW: Follow-up.

## Data Availability

Data may be available upon reasonable request.

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
