# Peer review of "Non-Surgical Periodontal Treatment Impact on Subgingival Microbiome and Intra-Oral Halitosis"

_ijms, 2023, doi:10.3390/ijms24032518_

Round 1

Reviewer 1 Report

As strong points, I would highlight the presentation of the baseline of the study subjects, the precision of the graphics and the striking colors used.
On the other hand, as it is a population with periodontitis, there could be patients with halitosis without periodontitis who present the same bacterial characteristics. In any case, it is understood that your research focuses on the response to periodontal treatment, which is why it is understood that there are no healthy controls.
  Some suggestions:
Figure 4 although it is very striking, is a bit difficult to read and interpret, perhaps it would be convenient to make a bar graph that clearly compares the before and after bacterial count highlighting the statistical significance results.

Reviewer 2 Report

thank you for submiting this paper

The research is quite interesting even if the number of patients seems quite limited for this kind of research

the major problem of your paper is the quality of the redaction, you really to ask some native english people to check the manuscript and improve the english (for example in the abstract line 28 referred for in place of referred to...).

concerning the methodology, the materials and methods is really good, nothing has to be changed. tHE ONLY LIMITATION IS THE NUMBER OF PATIENTS (20) which is a minimum 

Conclusions are interesting with a correlation of 2 perio pathogens with halitosis but these pathogens are not the ones classically linked to the severe forms of periodontitis 

Reviewer 3 Report

I thank the authors for presenting these scientific findings in a well written format. Here are my comments to make the manuscript further suitable for more wider variety of readers. 

1. Abstract: Please write the full form of VSC,  volatile sulphur compounds (VSC).

2. Introduction: There should be further information in the literature review section. More information should be added on the volatile sulphur compounds (VSC). Please state the litrature on this VSC levels for dentofacial research and also other information on the volatile organic compounds (VOC) in dental research. For example, this biosensor based study on segmenting different oral cancer cells should be cited to show the importance of studying volatile compounds (VSCs and VOCs) in detecting oral diseases. 

https://doi.org/10.1016/j.bios.2022.114814 
